# Usefulness of a TDM-Guided Approach for Optimizing Teicoplanin Exposure in the Treatment of Secondary Bloodstream Infections Caused by Glycopeptide-Susceptible *Enterococcus faecium*

**DOI:** 10.3390/microorganisms13010162

**Published:** 2025-01-14

**Authors:** Milo Gatti, Matteo Rinaldi, Maddalena Giannella, Pierluigi Viale, Federico Pea

**Affiliations:** 1Department of Medical and Surgical Sciences, Alma Mater Studiorum University of Bologna, 40138 Bologna, Italy; milo.gatti2@unibo.it (M.G.); matteo.rinaldi23@unibo.it (M.R.); maddalena.giannella@unibo.it (M.G.); pierluigi.viale@unibo.it (P.V.); 2Clinical Pharmacology Unit, Department for Integrated Infectious Risk Management, IRCCS Azienda Ospedaliero-Universitaria di Bologna, 40138 Bologna, Italy; 3Infectious Disease Unit, Department for Integrated Infectious Risk Management, IRCCS Azienda Ospedaliero-Universitaria of Bologna, 40138 Bologna, Italy

**Keywords:** teicoplanin, glycopeptide-susceptible *Enterococcus faecium*, bloodstream infection, TDM-guided ECPA strategy, microbiological eradication, source control

## Abstract

To assess the clinical usefulness of teicoplanin optimized by means of a therapeutic drug monitoring (TDM)-guided approach for treating secondary bloodstream infections (BSIs) caused by *Enterococcus faecium*. Hospitalized patients having in the period 1 March 2021–31 October 2024 a documented BSI caused by glycopeptide-susceptible *Enterococcus faecium* being treated with teicoplanin as definitive targeted therapy optimized by means of a real-time TDM-guided expert clinical pharmacological advice (ECPA) program were retrospectively included. Teicoplanin trough concentrations (C_min_) ranging from 20 to 30 mg/L were defined as the desired target of efficacy based on international guidelines. Univariate analysis was performed for assessing variables potentially associated with microbiological failure (defined as persistence at the infection site of the index *Enterococcus faecium* strain after more than 7 days from starting treatment as documented by follow-up blood cultures). Overall, 67 patients (median age 70 years; male 55.2%) were included. Catheter-related BSIs (50.7%) and intrabdominal/biliary tract (29.9%) infections were the main sources of *Enterococcus faecium* BSI. The desired target of teicoplanin C_min_ was attained in 62.7% of patients at the first TDM assessment and significantly increased to 85.1% (*p* = 0.003) at subsequent TDM-guided ECPA instances during the overall treatment course. Microbiological eradication was obtained in 95% of cases (63/67). In the univariate analysis, failing effective source control was the only variable associated with an increased risk of microbiological failure (75.0% vs. 12.7%; *p* = 0.01). Targeted TDM-guided teicoplanin therapy, coupled with effective source control of the primary infection site by granting microbiological eradication in the vast majority of cases, may be considered a reasonable strategy for managing glycopeptide-susceptible *Enterococcus faecium* secondary BSIs.

## 1. Introduction

*Enterococcus faecium* is a commensal microorganism of the intestinal tract that may cause several types of infections, including bloodstream infections (BSI) [1,2,3,4]. Specifically, the prevalence of *Enterococcus faecium*-related BSIs has been increasing worldwide and has reached high rates in recent years, namely around 10% of the overall hospital-detected BSIs [5,6,7]. *Enterococcus faecium*-related infections usually affect elderly and fragile patients having specific risk factors, namely being affected by different types of comorbidities (i.e., malignancy, diabetes, or solid organ transplantation, having hematopoietic stem cell transplantation), or having indwelling devices (central venous catheters) [8,9,10,11,12,13,14]. Although *Enterococcus faecium* BSIs are burdened by high mortality rates (i.e., 20–50%) [15,16], it is still unclear whether attributable mortality should be linked to the intrinsic virulence of the pathogen or to the underlying conditions of the patient.

Glycopeptide-susceptible *Enterococcus faecium* is resistant to beta-lactams and is considered a difficult-to-treat pathogen, so choosing the most appropriate treatment still remains a debated issue [2,8]. Based on clinical guidance, vancomycin is usually considered a first-line treatment, whereas linezolid and daptomycin may be potential alternatives [2,8]. In regard to the role of the two latter options in enterococcal BSIs, it should be recognized that studies documenting linezolid efficacy are quite few, and those concerning daptomycin efficacy are quite controversial due to high clinical failure rates [2,8,17,18].

Teicoplanin is a glycopeptide antibiotic that may represent a valuable alternative to vancomycin, as it has better in vitro activity against Enterococci and lowers nephrotoxicity risk in patients [19,20]. Considering the peculiar pharmacokinetic features, namely a very high plasma protein binding and a very long elimination half-life [20,21], a therapeutic drug monitoring (TDM)-guided strategy is recommended for attaining optimal trough concentrations (C_min_) of 20–30 mg/L with the most appropriate regimens based on a single daily or even a thrice weekly dose [22,23,24,25].

To date, the role of teicoplanin in the treatment of *Enterococcus faecium* BSIs was assessed in a few retrospective observational studies [26,27]. However, none of these assessed the usefulness of a TDM-guided approach in optimizing teicoplanin exposure nor tested whether any variable potentially associated with microbiological failure could exist.

The aim of this study was to assess the usefulness of a TDM-guided approach of teicoplanin in optimizing teicoplanin treatment of *Enterococcus faecium* BSIs and to test whether any variable potentially associated with microbiological failure could exist.

## 2. Materials and Methods

### 2.1. Study Design

This retrospective cohort study was carried out at the IRCCS Azienda Ospedaliero-Universitaria of Bologna, Italy. It included adult hospitalized patients having, in the period 1 March 2021–31 October 2024, a documented BSI caused by glycopeptide-susceptible *Enterococcus faecium* being treated with teicoplanin as definitive targeted therapy optimized by means of a real-time TDM-guided expert clinical pharmacological advice (ECPA) program. Only patients who underwent follow-up blood cultures after at least 48 h of targeted therapy with teicoplanin were included. Patients either not having follow-up blood cultures or having follow-up blood cultures in the first 48 h were excluded because the microbiological outcome could not be correctly assessed. Also, patients with polymicrobial BSI were excluded. A summary of the study design is reported in Figure 1. The study was approved by the local ethical committee (n. 442/2021/Oss/AOUBo approved on 28 June 2021) and conducted in agreement with the guidelines of the Declaration of Helsinki.

### 2.2. Data Collection

Demographic data (age, sex, height, weight, body mass index [BMI]), clinical/laboratory data (underlying diseases, Charlson Comorbidity Index [CCI] score [28], presence of immunosuppression, admission ward [medical, surgical, and/or intensive care unit], baseline creatinine clearance [CL_CR_] estimated by means of the CKD-EPI formula [29], need for continuous renal replacement therapy [CRRT] or intermittent hemodialysis [IHD], occurrence of augmented renal clearance [ARC], serum albumin), microbiological data (source of bacteremia, *Enterococcus faecium* susceptibility pattern, minimum inhibitory concentration [MIC] value for teicoplanin), treatment data (teicoplanin dosing regimen and plasma concentrations, treatment duration, achievement of effective source control), and outcome data (microbiological eradication, persistence of BSI, relapse of BSI, 30-day clinical cure, and 30-day mortality) were collected for included cases.

Immunosuppression was defined as the occurrence of one or more of the following conditions: long-term use of corticosteroids, immunosuppressants and/or of biologic and/or antineoplastic agents; or the presence of previous underlying solid organ (SOT), or hematopoietic stem cell transplantation (HSCT), or HIV disease or autoimmune disease [30].

ARC was defined as a normal serum creatinine level coupled with an estimated CL_CR_ > 130 mL/min/1.73 m^2^ in males and >120 mL/min/1.73 m^2^ in females [31].

Hypoalbuminemia was defined as the occurrence of serum albumin levels below 3.5 g/dL [32].

BSI was defined as the isolation of *Enterococcus faecium* from at least one of two blood cultures carried out from different sites according to the Centers for Disease Control and Prevention (CDC) criteria [33]. The source of bacteremia was classified as low-risk in the case of BSI deriving from the urinary tract, the vascular catheter, or the biliary tract, and high-risk for all other sites (namely the respiratory tract, skin, soft tissue, or endocardium) [34].

*Enterococcus faecium* strains were identified by means of the MALDI-ToF mass spectrometry using the Maldi Biotyper Sirius system (Bruker Daltonics, Bremen, Germany). Teicoplanin susceptibility was tested by means of a semi-automated broth microdilution method (Microscan Beckman NMDRM1, Beckman Coulter, Milan, Italy). MIC results were interpreted in agreement with the European Committee on Antimicrobial Susceptibility Testing (EUCAST) clinical breakpoints. Strains having a MIC value above the clinical breakpoint of 2 mg/L were defined as resistant [35].

Effective source control was defined as removal of the vascular catheter in case of catheter-related BSI [CR-BSI] or of the urinary stent in case of BSI secondary to urinary tract infection [UTI] drainage of abdominal/biliary abscesses, debridement of skin and soft tissue lesions in case of BSI secondary to skin and soft tissue infection [SSTI], or cardiac valve replacement in case of endocarditis during targeted therapy with teicoplanin [27].

### 2.3. Teicoplanin Dosing Regimen, Sampling Procedure, and TDM-Guided ECPA Program for Optimizing Teicoplanin Exposure

Targeted therapy with teicoplanin was prescribed by the treating physicians based on antimicrobial susceptibility tests. Therapy was always started with a loading dose (LD) period, namely 12 mg/kg every 12 h over 1 h for 5 doses, followed by a maintenance dose (MD) over 1 h infusion initially based on the patient’s estimated CL_CR_ (6 mg/kg q12h if CL_CR_ > 60 mL/min/1.73 m^2^; 50% of full dose if CL_CR_ ranged from 30 to 60 mL/min/1.73 m^2^; 25% of full dose if CL_CR_ < 30 mL/min/1.73 m^2^ or patients undergoing IHD) in agreement with international guidelines [36]. The LD was mandatory to promptly attain the desired teicoplanin exposure, and depending its calculation only on volume of distribution and not on clearance (LD = Ctarget × Vd), it was administered to all of the patients irrespective of renal function as previously recommended [22,23,24,25].

Subsequently, teicoplanin therapy was personalized in each single patient by means of a real-time TDM-guided ECPA program, as previously reported [37]. The first TDM-guided ECPA was performed after completing the LD period and reassessed every 48–72 h whenever feasible. Blood samples for measuring teicoplanin through plasma concentrations (C_min_) were collected 5–15 min before dosing. Teicoplanin C_min_ were promptly measured in real-time by means of a validated fluorescence polarization immunoassay method [37], so that TDM-guided ECPAs for dosing adjustment were provided to clinicians on the same day of blood sampling.

The optimal pharmacokinetic/pharmacodynamic (PK/PD) target of teicoplanin was defined as attaining a teicoplanin C_min_ ≥ 20 mg/L. This approach was valuable in attaining an area under a time-to-concentration curve (AUC)/MIC ratio of 500–900, as previously recommended [22,23,36,37].

### 2.4. Outcome Definition

Microbiological eradication or failure was defined as the eradication from or the persistence at the infection site of the index *Enterococcus faecium* strain as documented by follow-up blood cultures carried out after more than 7 days from starting teicoplanin treatment [34]. Persistence of *Enterococcus faecium* BSI was defined as the isolation of the index strain in follow-up blood cultures after >48 h of treatment, whereas relapse was defined as a positive blood culture to *Enterococcus faecium* after documented clearance in the first 30 days after the index culture [34]. Clinical cure was defined as the complete resolution of signs and symptoms of infection coupled with documented microbiological eradication at the end of treatment and absence of relapse at 30-day follow-up and/or of attributable mortality due to *Enterococcus faecium* infection [34].

### 2.5. Statistical Analysis

Continuous data are described as median and interquartile range (IQR), whereas categorical variables are expressed as counts and percentages. Univariate analysis comparing patients having microbiological eradication vs. microbiological failure was performed by means of the Fisher’s exact test, the χ^2^ test (for categorical variables,) or the Mann–Whitney U test (for continuous variables). The model was adjusted for age and gender in order to minimize the risk of potential confounders. Independent covariates having a *p* value < 0.20 at the univariate analysis were included in the multivariate logistic regression model. Statistical significance was defined as a *p* value < 0.05. Statistical analysis was performed using MedCalc for Windows (MedCalc statistical software, version 19.6.1, MedCalc Software Ltd., Ostend, Belgium).

## 3. Results

Overall, a total of 67 hospitalized patients received teicoplanin monotherapy optimized by means of a TDM-guided strategy for managing glycopeptide-susceptible *Enterococcus faecium* BSIs during the study period. The demographic and clinical features of the included patients are summarized in Table 1.

The median (IQR) age was 70 years olf (60–77 years), with a slight male preponderance (55.2%). The median (IQR) CCI was 6 points (4–8 points), and 52.2% of patients were immunodepressed. Most cases were admitted to medical wards (44/67; 65.7%), and 11 (16.4%) required ICU admission.

The median (IQR) baseline CLCr was 62 mL/min/1.73 m^2^ (35.5–87.5 mL/min/1.73 m^2^). Three patients each (4.5%) underwent CRRT/IHD or experienced ARC. Hypoalbuminemia was documented in five cases (10.4%).

Most of the *Enterococcus faecium* secondary BSIs were CR-BSIs (34/67; 50.7%), followed by those secondary to IAI (20/67; 29.9%) and UTI (8/67; 11.9%). Endocarditis and BSI associated with SSTI were reported in three (4.5%) and two cases (2.0%), respectively. Most patients (56/67; 83.6%) had effective source control. Most isolates had a teicoplanin MIC value of 1 mg/L (62/67; 92.5%). Four strains had a MIC value of 2 mg/L. All *Enterococcus faecium* strains were also susceptible to linezolid, and all but one were susceptible to daptomycin.

Teicoplanin was administered at a median (IQR) daily dose of 600 mg (400 mg–800 mg) with a median (IQR) treatment duration of 12 days (9–15.5 days). Median (IQR) teicoplanin C_min_ at first TDM assessment was 22 mg/L (15 mg/L–33 mg/L), whereas the median (IQR) average teicoplanin C_min_ during the overall treatment course was 26 mg/L (21.6 mg/L–29.8 mg/L).

A total of 229 TDM-based ECPAs were performed for optimizing the teicoplanin dosing regimen, with a median (IQR) number of 3 (2–4) per patient. At the first TDM-based ECPA, a dosing reduction was recommended in the majority of cases (41/67; 61.1%), whereas a dosing increase was recommended in six patients (9.0%). Overall, dosing adjustments were recommended in 93 out of 229 TDM-based ECPAs (40.6%), which included an increase of 13 (5.7%) and a decrease of 80 (34.9%). Optimal teicoplanin exposure was attained in 62.7% of patients at first TDM assessment, and using the TDM-guided approach significantly increased this to 85.1% of overall cases during the overall treatment course (*p* = 0.003).

Microbiological eradication was obtained and reported in 63 out of 67 cases (94.0%), whereas failure occurred in four cases (6.0%; two persisting BSI and 30-day relapse each). Resistance to teicoplanin with detection of glycopeptide-resistant *Enterococcus faecium* occurred at follow-up blood cultures only in one patient (1.5%). Clinical cure was documented in 79.1% of patients (53/67), and the 30-day mortality rate was 11.9%.

Univariate analysis assessing potential variables potentially associated with microbiological eradication vs. failure is reported in Table 2.

Only failing effective source control was significantly associated with an increased risk of microbiological failure (75.0% vs. 12.7%; *p* = 0.01).

## 4. Discussion

To the best of our knowledge, this is the first study assessing the usefulness of a TDM-guided strategy of teicoplanin therapy for optimizing the treatment of *Enterococcus faecium* secondary BSIs. Notably, our findings showed that this approach allowed a significant increase in the proportion of optimal teicoplanin PK/PD target attainment compared to the first TDM assessment, ensuring microbiological eradication and favorable clinical outcomes in the vast majority of patients. Microbiological failure was limited to a minority of cases, for which a lack of effective primary source control was the only independent risk factor associated.

The identity of the best therapeutic option for treating glycopeptide-susceptible *Enterococcus faecium* BSIs still represents an unmet clinical need nowadays [8]. A propensity score-matched comparative study including 105 patients affected by glycopeptide-susceptible BSIs found no difference between linezolid and glycopeptides in terms of clinical cure rate (57.1% vs. 46.4%; *p* = 0.59), microbiological eradication rate (78.6% vs. 71.4%; *p* = 0.76), 30-day mortality rate (28.6% vs. 42.9%; *p* = 0.403), and treatment discontinuation due to adverse events (11.5% vs. 18.5%; *p* = 0.70) [34]. Likewise, recent studies found no significant difference in terms of efficacy between teicoplanin and vancomycin in the management of glycopeptide-susceptible *Enterococcus faecium* BSIs, with advantages in terms of better safety profile for teicoplanin [26,27]. This is in agreement with the valuable role that teicoplanin showed in our study in terms of both good microbiological eradication rate and clinical rate in this scenario. Specifically, a retrospective analysis of a multicenter national prospective study including 97 patients affected by glycopeptide-susceptible *Enterococcus faecium* BSIs (33 treated with teicoplanin and 64 receiving vancomycin) found no significant difference among groups both in terms of 30-day in-hospital mortality rate (18.2% vs. 26.6%; *p* = 0.36) and of 7-day mortality (6.1% vs. 15.6%; *p* = 0.21) [26]. Additionally, teicoplanin was not found in the multivariate analysis to be an independent predictor of mortality (aOR 0.72; 95% CI 0.28–1.86; *p* = 0.49) [26]. Similarly, a propensity score-adjusted retrospective comparative study including 164 patients affected by glycopeptide-susceptible *Enterococcus faecium* BSIs (74 treated with teicoplanin and 90 receiving vancomycin) found non-inferiority of teicoplanin compared to vancomycin in terms of clinical success at the end of treatment, with an adjusted absolute difference in effectiveness of 9.9% (95% CI −0.9–20.0%; *p* = 0.07) [27]. Interestingly, the risk of acute kidney injury was significantly lower with teicoplanin (OR 0.24; 95% CI 0.07–0.86; *p* = 0.02) [27].

Of note, none of the previous studies assessed the role that a real-time TDM-guided strategy may have in optimizing teicoplanin exposure in the management of glycopeptide-susceptible *Enterococcus faecium* BSIs. We found that slightly more than 60% of the included patients attained an optimal teicoplanin PK/PD target after the loading period, consistent with what was reported by Yamaguchi et al. in their analysis [27]. However, adopting a real-time TDM-guided ECPA approach allowed us to significantly increase the proportion of optimal exposure during the treatment course, up to more than 85% of patients. Suboptimal teicoplanin PK/PD target attainment did not emerge as an independent predictor of microbiological failure in our analysis, and the potential impact of this on the outcome of *Enterococcus faecium* BSIs should be further assessed in future dedicated studies.

Failure in achieving a complete and effective source control emerged as the only significant predictor of microbiological failure in our analysis. This finding was consistent with previous studies showing that catheter removal had a favorable clinical impact in the case of Enterococcal CR-BSIs regardless of whether antibiotic therapy was appropriate [38,39,40,41]. Although international guidelines recommended the adoption of device removal coupled with targeted antibiotic therapy in the case of Enterococcal CR-BSIs, no distinction was carried out between *Enterococcus faecalis* and *Enterococcus faecium* [42]. Based on these findings, achieving effective and complete source control should be considered mandatory in patients affected by *Enterococcus faecium* BSIs for minimizing the risk of occurrence of persisting and/or relapsing BSI.

Overall, our findings suggest that adopting a TDM-guided strategy for maximizing teicoplanin PK/PD target attainment was successful in obtaining microbiological eradication in a large proportion of patients with glycopeptide-susceptible *Enterococcus faecium* BSIs, being microbiological failure limited to cases failing effective source control.

The limitations of our study should be acknowledged. The study design was retrospective and monocentric, with a limited sample size. Multivariate analysis could not be performed due to the low number of patients having microbiological failure. This precluded us from investigating potential independent predictors associated with persisting and/or relapsing *Enterococcus faecium* BSIs. We also recognize that performing accurate sample size calculations for establishing statistical power would have added more value to the study. Unfortunately, we did not have a clear benchmark for sample size calculation since, to the best of our knowledge, ours was the first study exploring the usefulness of a TDM-guided approach in optimizing teicoplanin treatment of *Enterococcus faecium* BSIs and assessing potential variables associated with microbiological failure. However, we believe that the total number of patients included in our study (i.e., 67) could be enough for inferring reliable conclusions since it was quite similar to those included in two previous studies evaluating the role of teicoplanin in the management of *Enterococcus faecium* BSIs, namely 33 and 74 [26,27].

## 5. Conclusions

In conclusion, our findings suggest that targeting TDM-guided teicoplanin therapy coupled with achieving effective source control of the primary infection site, by granting microbiological eradication in the vast majority of cases, may be considered a reasonable and valuable strategy for properly managing glycopeptide-susceptible *Enterococcus faecium* secondary BSIs. Larger definitive prospective studies are warranted for confirming our contention and for definitively testing whether or not suboptimal teicoplanin exposure might be associated with microbiological failure in this very challenging scenario.

## Figures and Tables

**Figure 1 microorganisms-13-00162-f001:**
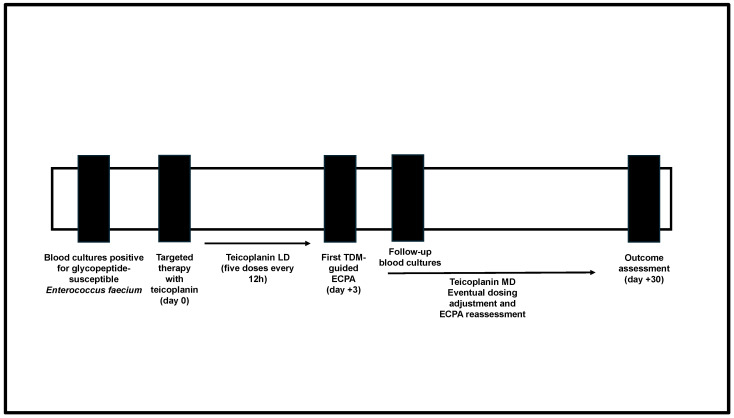
Study design flowchart. ECPA: expert clinical pharmacological advice; LD: loading dose; MD: maintenance dose; TDM: therapeutic drug monitoring.

**Table 1 microorganisms-13-00162-t001:** Demographic and clinical characteristics of the included patients having definitive monotherapy with TDM-guided teicoplanin for treating BSI caused by vancomycin-susceptible *Enterococcus faecium*.

Demographic and Clinical Variables	Patients (N = 67)
Patient demographics	
Age (years) [median (IQR)]	70 (60–77)
Gender (male/female) [n (%)]	37/30 (55.2/44.8)
Body weight (kg) [median (IQR)]	65.0 (60.0–75.5)
Body mass index (kg/m^2^) [median (IQR)]	23.0 (20.9–25.0)
Obesity [n (%)]	7 (10.4)
Admission ward [n (%)]	
Medical	44 (65.7)
Surgical	9 (13.4)
ICU	11 (16.4)
Hematology	3 (4.5)
Underlying conditions	
Charlson Comorbidity Index [median (IQR)]	6 (4–8)
Immunosuppression [n (%)]	35 (52.2)
Status of renal function and serum albumin levels	
Baseline CL_CR_ (mL/min/1.73 m^2^) [median (IQR)]	62.0 (35.5–87.5)
IHD/CRRT [n (%)]	3 (4.5)
Augmented renal clearance [n (%)]	3 (4.5)
Serum albumin (mg/dL) [median (IQR)]	2.95 (2.72–3.19)
Hypoalbuminemia [n (%)] *	5 (10.4)
Primary source of BSI	
CR-BSI	34 (50.7)
IAI/biliary	20 (29.9)
UTI	8 (11.9)
Endocarditis	3 (4.5)
SSTI	2 (3.0)
Failing in effective source control [n (%)]	11 (16.4)
*Enterococcus faecium* susceptibility	
Vancomycin	67 (100.0)
Linezolid	67 (100.0)
Daptomycin	66 (98.5)
Teicoplanin MIC	
0.5 mg/L	1 (1.5)
1 mg/L	62 (92.5)
2 mg/L	4 (6.0)
Teicoplanin treatment	
Daily dose (mg) [median (IQR)]	600 mg/day (400–800 mg/day)
Treatment duration (days) [median (IQR)]	12.0 (9.0–15.5)
Teicoplanin average C_min_ (mg/L) [median (IQR)]	26.0 (21.6–29.8)
Teicoplanin C_min_ at first TDM assessment [median (IQR)]	22.0 (15.0–33.0)
PK/PD target attainment	
Overall optimal PK/PD target [n (%)]	57 (85.1)
Overall suboptimal PK/PD target [n (%)]	10 (14.9)
Overall optimal PK/PD target at first TDM assessment [n (%)]	42 (62.7)
Overall suboptimal PK/PD target at first TDM assessment [n (%)]	25 (37.3)
ECPA program	
Overall TDM-based ECPAs	229
N. of TDM-based ECPA per treatment course [median (IQR)]	3 (2–4)
N. of dosage confirmations at first TDM assessment [n (%)]	20 (29.9)
N. of dosage decreases at first TDM assessment [n (%)]	41 (61.1)
N. of dosage increases at first TDM assessment [n (%)]	6 (9.0)
Overall n. of dosage confirmations [n (%)]	136 (59.4)
Overall n. of dosage decreases [n (%)]	80 (34.9)
Overall n. of dosage increases [n (%)]	13 (5.7)
Outcome	
Microbiological eradication [n (%)]	63 (94.0)
Resistance development [n (%)]	1 (1.5)
Persistent BSI	2 (3.0)
30-day relapse [n (%)]	2 (3.0)
Clinical cure [n (%)]	53 (79.1)
30-day mortality [n (%)]	8 (11.9)

* Serum albumin levels were available for 48 out of 67 patients. BSI: bloodstream infection; CL_CR_: creatinine clearance; C_min_: trough concentrations; CR-BSI: catheter-related bloodstream infection; CRRT: continuous renal replacement therapy; ECPA: expert clinical pharmacological advice; IAI: intrabdominal infection; ICU: intensive care unit; IHD: intermittent hemodialysis; IQR: interquartile range; MIC: minimum inhibitory concentration; PK/PD: pharmacokinetic/pharmacodynamic; SSTI: skin and soft tissue infection; TDM: therapeutic drug monitoring; UTI: urinary tract infection.

**Table 2 microorganisms-13-00162-t002:** Univariate and multivariate analyses comparing patients receiving teicoplanin for treating BSI caused by glycopeptide-susceptible *Enterococcus faecium* and showing microbiological eradication vs. microbiological failure.

Variables	Microbiological Eradication (N = 63)	Microbiological Failure (N = 4)	Univariate Analysis *p* Value
Patient demographics			
Age (years) [median (IQR)]	68.0 (59.0–77.0)	75.0 (69.5–81.8)	0.20
Gender (male/female) [n (%)]	35/28 (55.6/44.4)	2/2 (50.0/50.0)	0.99
Body weight (kg) [median (IQR)]	65.0 (60.0–76.0)	65.0 (58.8–72.1)	0.91
Body mass index (kg/m^2^) [median (IQR)]	23.0 (20.8–25.0)	25.0 (22.9–27.7)	0.27
Obesity [n (%)]	6 (9.5)	1 (25.0)	0.36
Admission ward [n (%)]			
Medical	41 (65.0)	3 (75.0)	0.99
Surgical	9 (14.3)	0 (0.0)	0.99
ICU	10 (15.9)	1 (25.0)	0.52
Hematology	3 (4.8)	0 (0.0)	0.99
Underlying conditions			
Charlson Comorbidity Index [median (IQR)]	6 (4–8)	7 (4.75–9.25)	0.47
Immunosuppression [n (%)]	32 (50.8)	3 (75.0)	0.62
Status of renal function and serum albumin levels			
Baseline CL_CR_ (mL/min/1.73 m^2^) [median (IQR)]	63.0 (37.0–89.0)	43.0 (22.5–66.0)	0.27
IHD/CRRT [n (%)]	3 (4.8)	0 (0.0)	0.99
Augmented renal clearance [n (%)]	3 (4.8)	0 (0.0)	0.99
Hypoalbuminemia [n (%)] *	4 (8.7)	1 (50.0)	0.20
Source of BSI [n (%)]			
CR-BSI	31 (49.2)	3 (75.0)	0.61
IAI/biliary	19 (30.1)	1 (25.0)	0.99
UTI	8 (12.7)	0 (0.0)	0.99
Endocarditis	3 (4.8)	0 (0.0)	0.99
SSTI	2 (3.2)	0 (0.0)	0.99
Failing effective source control [n (%)]	8 (12.7)	3 (75.0)	0.01
MIC value [n (%)]			
0.5 mg/L	1 (1.6)	0 (0.0)	0.99
1 mg/L	59 (93.7)	3 (75.0)	0.27
2 mg/L	3 (4.7)	1 (25.0)	0.22
Teicoplanin treatment and PK/PD target attainment			
Treatment duration (days) [median (IQR)]	12.0 (9.0–15.0)	17.5 (13.75–22)	0.08
Teicoplanin average C_min_ (mg/L) [median (IQR)]	26.0 (21.6–30.0)	26.9 (23.9–27.3)	0.67
Teicoplanin C_min_ at first TDM assessment [median (IQR)]	22.0 (15.0–32.0)	27.5 (19.3–40.5)	0.63
Overall teicoplanin suboptimal PK/PD target attainment	9 (14.3)	1 (25.0)	0.48
Overall teicoplanin suboptimal PK/PD target attainment at first TDM assessment	24 (38.1)	1 (25.0)	0.99

* Available for 46/63 and 2/4 patients in microbiological eradication and failure groups, respectively. BSI: bloodstream infection; CL_CR_: creatinine clearance; C_min_: trough concentrations; CR-BSI: catheter-related bloodstream infection; CRRT: continuous renal replacement therapy; IAI: intrabdominal infection; ICU: intensive care unit; IHD: intermittent hemodialysis; IQR: interquartile range; MIC: minimum inhibitory concentration; PK/PD: pharmacokinetic/pharmacodynamic; SSTI: skin and soft tissue infection; TDM: therapeutic drug monitoring; UTI: urinary tract infection.

## Data Availability

The data presented in this study are available on request from the corresponding author. The data are not publicly available due to privacy concerns.

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
