# Peer review of "Usefulness of a TDM-Guided Approach for Optimizing Teicoplanin Exposure in the Treatment of Secondary Bloodstream Infections Caused by Glycopeptide-Susceptible Enterococcus faecium"

_microorganisms, 2025, doi:10.3390/microorganisms13010162_

Round 1
Reviewer 1 Report
Comments and Suggestions for Authors
This study aims to To assess the clinical usefulness of teicoplanin optimized according to a therapeutic drug monitoring (TDM)-guided approach for the management of Enterococcus faecium secondary bloodstream infections (BSIs). It is well written and easy to read. Minor comments should be considered as follows:
1. Abstract: it should be one paragraph without division.
2. Intro: the aim of this study should be clearly described.
3. The author should a scheme to present the procedures and their flow in this study.
4. Is the sample size convenient for this study?
5. The authors should add limitations of this study, including sample size and analysis, in addition to future perspectives.
6. Conclusion: it is too short, please add significant findings.
Author Response
RESPONSE TO REVIEWERS
Manuscript ID: microorganisms-3407014 “Usefulness of a TDM-guided approach for optimizing teicoplanin exposure in the treatment of secondary bloodstream infections caused by glycopeptide-susceptible Enterococcus faecium” by Gatti et al.
Dear Editor,
We would like to thank you for the opportunity to resubmit a revised version of this manuscript. We appreciated the reviewer’s constructive comments. All have been carefully considered and incorporated, where and whenever possible, in the revision. Furthermore, as suggested we carefully reviewed the English in order to improve the readability, and the manuscript was carefully revised for reducing the similarity index. For this latter point, it should be noticed that different sentences reported as similarity (e.g., table legends, ethical committee approval, variable/outcome definitions) concern aspects that are commonly retrieved in studies with similar topic and design.
Our point-by-point responses are provided below.
Q= QUERY; A= ANSWER
Reviewer #1:
This study aims to to assess the clinical usefulness of teicoplanin optimized according to a therapeutic drug monitoring (TDM)-guided approach for the management of Enterococcus faecium secondary bloodstream infections (BSIs). It is well written and easy to read. Minor comments should be considered as follows:
We thank the reviewer for appreciating our manuscript.
Q1. Abstract: it should be one paragraph without division.
A1. Thank you for this suggestion. We removed in the Abstract the division in paragraphs.
Q2. Intro: the aim of this study should be clearly described.
A2. We thank the reviewer for this comment, allowing us to better clarify this important issue. We better described the aim of our study in the Introduction section (refer to Line 73-75).
Q3. The author should a scheme to present the procedures and their flow in this study.
A3. We thank the reviewer for this useful suggestion. We added in the Methods section a specific figure (Figure 1) reporting a dedicated flowchart of study procedure.
Q4. Is the sample size convenient for this study?
A4. We thank the reviewer for this relevant comment. Unfortunately, we did not have a clear benchmark for sample size calculation since, to the best of our knowledge, ours was the first study exploring the usefulness of a TDM-guided approach in optimizing teicoplanin treatment of Enterococcus faecium BSIs and assessing potential variables associated with microbiological failure. We believe that the total number of patients included in our study (i.e., 67) could be enough for inferring reliable conclusions since it was quite similar to those included in two previous studies evaluating the role of teicoplanin in the management of Enterococcus faecium BSIs, namely 33 and 74 (i.e., references 26-27). Additionally, by hypothesizing that the same principles reported in doi: 10.3390/antibiotics12121736 for calculating the sample size needed for testing a 50% microbiological eradication rate increase with optimal PK/PD target attainment of beta-lactams could be reliably applied to teicoplanin, we calculated that a lower total number of patients than included in our cohort, namely 44, could be appropriate with a type-I error rate of 5% and a power level of 80%. Besides, we recognize that performing accurate sample size calculation for establishing statistical power would have added more value to the study, so that this lack was acknowledged as a limit of our study (refer to Discussion section; Line 302-311).
Q5. The authors should add limitations of this study, including sample size and analysis, in addition to future perspectives.
A5. We thank the reviewer for this relevant comment. We extended the paragraph concerning limitations of our study including specific considerations on sample size and statistical analysis as suggested (refer to Discussion section; Line 298-311).
Q6. Conclusion: it is too short, please add significant findings.
A6. We thank the reviewer for this suggestion. We extended the Conclusion section by adding significant findings of our study (refer to Line 313-319).
Reviewer 2 Report
Comments and Suggestions for Authors
Dear Authors, thank you for submitting your manuscript which topic is very interesting due to the difficulties in treating BSI due to Enterococcus species. The use of a specific antimicrobial treatment (teicoplanin) applying TDM is doubly interesting because opens the door to alternative molecules in the treatment of bloodstream infection caused by Enterococcus sensitive to glycopeptide strains and to their use in a targeted way avoiding the increase of AMR.
Remarkable is the pointing of the Authors about the limits of their research.
The quality of the writing is good and the English also is very good.
Some comments:
1- in lines 75-77 is written "Only patients who underwent follow-up blood cultures after at least 48 hours of targeted therapy with teicoplanin were included.". And the others? I presume that all the patients were hospitalized. Please explain this sentence
2- in lines 104-105 is written "and high-risk for all other sites". Please, Could you specify the sites for high-risk BSI?
3- line 132 "Teicoplanin Cmin were measured by means of a validated fluorescence polarization immunoassay method". How long is required for teicoplanin trough concentration dosing?
4- line 205 "...the 30-day mortality rate was 11.9%." This percentage is quite high. What did it cause the 30-day mortality? Please specify
Author Response
RESPONSE TO REVIEWERS
Manuscript ID: microorganisms-3407014 “Usefulness of a TDM-guided approach for optimizing teicoplanin exposure in the treatment of secondary bloodstream infections caused by glycopeptide-susceptible Enterococcus faecium” by Gatti et al.
Dear Editor,
We would like to thank you for the opportunity to resubmit a revised version of this manuscript. We appreciated the reviewer’s constructive comments. All have been carefully considered and incorporated, where and whenever possible, in the revision. Furthermore, as suggested we carefully reviewed the English in order to improve the readability, and the manuscript was carefully revised for reducing the similarity index. For this latter point, it should be noticed that different sentences reported as similarity (e.g., table legends, ethical committee approval, variable/outcome definitions) concern aspects that are commonly retrieved in studies with similar topic and design.
Our point-by-point responses are provided below.
Q= QUERY; A= ANSWER
Reviewer #2:
Dear Authors, thank you for submitting your manuscript which topic is very interesting due to the difficulties in treating BSI due to Enterococcus species. The use of a specific antimicrobial treatment (teicoplanin) applying TDM is doubly interesting because opens the door to alternative molecules in the treatment of bloodstream infection caused by Enterococcus sensitive to glycopeptide strains and to their use in a targeted way avoiding the increase of AMR.
Remarkable is the pointing of the Authors about the limits of their research.
The quality of the writing is good and the English also is very good.
We thank the reviewer for appreciating our study.
Some comments:
Q1. - in lines 75-77 is written "Only patients who underwent follow-up blood cultures after at least 48 hours of targeted therapy with teicoplanin were included.". And the others? I presume that all the patients were hospitalized. Please explain this sentence
A1. We thank the reviewer for this comment, allowing us to better clarify this point. Patients who didn’t performed follow-up blood cultures after at least 48 hours of targeted therapy with teicoplanin were excluded because microbiological outcome was not assessable in these cases. We better explained this issue in the Methods section (refer to Line 84-86).
Q2. - in lines 104-105 is written "and high-risk for all other sites". Please, Could you specify the sites for high-risk BSI?
A2. Thank you for this suggestion and for allowing us to better clarify the source of high-risk BSI, namely those deriving from respiratory tract, skin and soft tissue, or endocarditis. We detailed this issue in the Methods section (refer to Line 120).
Q3. - line 132 "Teicoplanin Cmin were measured by means of a validated fluorescence polarization immunoassay method". How long is required for teicoplanin trough concentration dosing?
A3. We thank the reviewer for this relevant comment, allowing us to better clarify this issue. Teicoplanin trough concentrations were available in the same day in which the samples were collected and dosing adjustment was performed in real-time. The median turnaround time was approximatively 8 hours as previously reported (reference no. 37). We detailed this issue in the Methods section (refer to Line 151-164).
Q4. - line 205 "...the 30-day mortality rate was 11.9%." This percentage is quite high. What did it cause the 30-day mortality? Please specify
A4. We thank the reviewer for this comment, allowing us to better clarify this point. Although the 30-day mortality rate could seem quite high at first glance, it is noteworthy that our mortality rate is significantly lower compared to those reported in literature (i.e., 20-50%; reference no. 15-16), and that the large amount of this prevalence is probably attributable to underlying conditions of included patients, as detailed in the Introduction section (refer to Line 49-52).
Reviewer 3 Report
Comments and Suggestions for Authors
Minor Comments
Line 17-22: How does the TDM-guided approach ensure optimal teicoplanin trough concentrations for treating bloodstream infections caused by glycopeptide-susceptible Enterococcus faecium?
Line 49-55: Why is vancomycin considered the first-line treatment for glycopeptide-susceptible Enterococcus faecium infections, and what are the limitations of alternative treatments like daptomycin and linezolid?
Line 64-66: What are the gaps in existing research regarding teicoplanin’s role in treating Enterococcus faecium bloodstream infections, especially concerning therapeutic drug monitoring?
Line 122-126: How does the initial loading dose regimen of teicoplanin vary based on renal function, and how effective is this approach for achieving therapeutic concentrations?
Line 201-205: What factors contribute to the observed microbiological eradication rate of 94% with TDM-guided teicoplanin therapy, and how does this compare to failure rates?
Line 217-218: Why is failing effective source control identified as a significant risk factor for microbiological failure, and how can this be addressed in clinical practice?
Line 251-257: What is the impact of a real-time TDM-guided strategy on achieving optimal teicoplanin PK/PD targets, and how does it compare to non-guided approaches in clinical outcomes?
Comments on the Quality of English LanguageMinor Comments
Author Response
RESPONSE TO REVIEWERS
Manuscript ID: microorganisms-3407014 “Usefulness of a TDM-guided approach for optimizing teicoplanin exposure in the treatment of secondary bloodstream infections caused by glycopeptide-susceptible Enterococcus faecium” by Gatti et al.
Dear Editor,
We would like to thank you for the opportunity to resubmit a revised version of this manuscript. We appreciated the reviewer’s constructive comments. All have been carefully considered and incorporated, where and whenever possible, in the revision. Furthermore, as suggested we carefully reviewed the English in order to improve the readability, and the manuscript was carefully revised for reducing the similarity index. For this latter point, it should be noticed that different sentences reported as similarity (e.g., table legends, ethical committee approval, variable/outcome definitions) concern aspects that are commonly retrieved in studies with similar topic and design.
Our point-by-point responses are provided below.
Q= QUERY; A= ANSWER
Reviewer #3:
Q1. Line 17-22: How does the TDM-guided approach ensure optimal teicoplanin trough concentrations for treating bloodstream infections caused by glycopeptide-susceptible Enterococcus faecium?
A1. We thank the reviewer for this comment, allowing us to better clarify this specific point. Indeed, several international guidance and guidelines (references no. 22-25) strongly recommended the adoption of a TDM-guided approach for maximizing the attainment of optimal PK/PD targets with glycopeptides including teicoplanin, considering its PK properties (namely high plasma protein binding and long half-life), as detailed in the Introduction section (refer to Line 62-66). We added a brief clarification on this issue also in the Abstract (refer to Line 22-23).
Q2. Line 49-55: Why is vancomycin considered the first-line treatment for glycopeptide-susceptible Enterococcus faecium infections, and what are the limitations of alternative treatments like daptomycin and linezolid?
A2. We thank the reviewer for this comment, allowing us to better clarify this issue. Vancomycin is considered the first-line treatment in this scenario according to studies documenting its valuable clinical efficacy and favorable PK/PD profile in case of bloodstream infections, as reported in clinical guidance (reference no. 8). On the other side, the evidence for linezolid efficacy in Enterococcus faecium BSIs are quite few (reference no. 17), whereas as regards daptomycin a non-negligible rate of clinical failure was reported in previous studies (reference no. 18). We better specified these points in the Introduction section (refer to Line 55-59).
Q3. Line 64-66: What are the gaps in existing research regarding teicoplanin’s role in treating Enterococcus faecium bloodstream infections, especially concerning therapeutic drug monitoring?
A3. We thank the reviewer for this suggestion, allowing to better clarify the existing gaps regarding the role of teicoplanin in the management of Enterococcus faecium BSIs. Specifically, to the best of our knowledge, only two observational retrospective studies (reference no. 26-27) have previously assessed the clinical efficacy of teicoplanin in the management of this infection. However, in none of these studies the role of a TDM-guided approach in personalizing teicoplanin dosing regimens for maximizing PK/PD target attainment was assessed, as well as none of these studies have investigated potential variables associated with microbiological failure in terms of persistence and/or relapse of bacteremia. We better detailed these issues in the Introduction section (refer to Line 67-71).
Q4. Line 122-126: How does the initial loading dose regimen of teicoplanin vary based on renal function, and how effective is this approach for achieving therapeutic concentrations?
A4. We thank the reviewer for this comment. Clinical guidance and guidelines and available literature (references no. 22-25) strongly support the administration of teicoplanin high loading dose regimens for maximizing the attainment of therapeutically effective Cmin irrespective of the degree of renal function in 60-85% of patients in different clinical scenarios (reference no. 23). Conversely, teicoplanin maintenance dosing should be modified according to renal function as recommended by clinical guidance and guidelines (references no. 22-25), considering the predominant renal elimination exhibited by this agent. We better detailed this issue in the Methods section (refer to Line 143-146).
Q5. Line 201-205: What factors contribute to the observed microbiological eradication rate of 94% with TDM-guided teicoplanin therapy, and how does this compare to failure rates?
A5. We thank the reviewer for this comment, allowing us to better clarify this point. As detailed in Table 1 and 2, the observed microbiological eradication rate of 94% could be associated with the fact that more than 85% of included patients attained an optimal teicoplanin PK/PD target by means of the implementation of a TDM-guided approach, and that failure in achieving an effective and complete source control was limited to less than 20% of cases. Indeed, as regards to the comparison between patients showing microbiological eradication vs. failure (Table 2), only failing effective source control was associated with higher risk of microbiological failure. These issues were detailed and discussed in the Discussion section (refer to Line 273-293).
Q6. Line 217-218: Why is failing effective source control identified as a significant risk factor for microbiological failure, and how can this be addressed in clinical practice?
A6. We thank the reviewer for this comment, allowing us to better clarify this relevant issue. As already discussed (refer to Line 284-293), the finding of a significantly higher risk of microbiological failure in patients failing to have complete and effective source control was consistent with previous studies showing that catheter removal had a favorable clinical impact in case of Enterococcal catheter-related bloodstream infections. Additionally, also international guidelines strongly recommended device removal coupled with targeted antibiotic therapy in case of Enterococcal CR-BSIs, thus further underlying the relevance of achieving an effective source control. We discussed better potential impact on clinical practice of this relevant finding in the Discussion section (refer to Line 290-293).
Q7. Line 251-257: What is the impact of a real-time TDM-guided strategy on achieving optimal teicoplanin PK/PD targets, and how does it compare to non-guided approaches in clinical outcomes?
A7. We thank the reviewer for this comment, allowing us to better clarify this point. As reported in the Results section (refer to Line 219-221) and further discussed (refer to Line 273-283), the adoption of a TDM-guided strategy allowed to significantly increase the attainment of optimal teicoplanin PK/PD target compared to first TDM assessment (85.1% vs. 62.7%; p=0.003). A comparison with a standard approach not including TDM-guided dosing optimization was not possible in our analysis considering that all including patients underwent a real-time TDM-guided ECPA strategy, as specified in the Methods section (refer to Line 81-83). This aspect may be investigated in future dedicated studies, as suggested in the Discussion section (refer to Line 280-283 and 317-319).
Reviewer 4 Report
Comments and Suggestions for Authors
The article is well-written and needs of the current era. The authors presented valuable findings in managing E. faecium infections using teicoplanin. However, I have only minor comments for the authors.
Line 21: Please correct the spelling "through".
Line 24: The term "microbiological failure" needs to be clarified or changed with suitable words.
Line 42: Can you mention at which rate?
Author Response
RESPONSE TO REVIEWERS
Manuscript ID: microorganisms-3407014 “Usefulness of a TDM-guided approach for optimizing teicoplanin exposure in the treatment of secondary bloodstream infections caused by glycopeptide-susceptible Enterococcus faecium” by Gatti et al.
Dear Editor,
We would like to thank you for the opportunity to resubmit a revised version of this manuscript. We appreciated the reviewer’s constructive comments. All have been carefully considered and incorporated, where and whenever possible, in the revision. Furthermore, as suggested we carefully reviewed the English in order to improve the readability, and the manuscript was carefully revised for reducing the similarity index. For this latter point, it should be noticed that different sentences reported as similarity (e.g., table legends, ethical committee approval, variable/outcome definitions) concern aspects that are commonly retrieved in studies with similar topic and design.
Our point-by-point responses are provided below.
Q= QUERY; A= ANSWER
Reviewer #4:
The article is well-written and needs of the current era. The authors presented valuable findings in managing E. faecium infections using teicoplanin. However, I have only minor comments for the authors.
We thank the reviewer for appreciating our paper.
Q1. Line 21: Please correct the spelling "through".
A1. Thank you for this suggestion. However, a misunderstanding could be occurred, considering that the correct spelling in this context is “trough” and not “through” for meaning the assessment of teicoplanin concentrations before the administration of the scheduled next dose.
Q2. Line 24: The term "microbiological failure" needs to be clarified or changed with suitable words.
A2. We thank the reviewer for this comment. We clarify in the Abstract (refer to Line 24-26) the term “microbiological failure” by providing the adopted definition.
Q3. Line 42: Can you mention at which rate?
A3. Thank you for this suggestion. We specified the rate of Enterococcus faecium bacteremia in the Introduction section (refer to Line 45).